# Effects of lifelong testosterone exposure on health and disease using Mendelian randomization

Pedrum Mohammadi-Shemirani[1,2,3], Michael Chong[1,2,4], Marie Pigeyre[1,5], Robert W Morton[6], Hertzel C Gerstein[1,5], Guillaume Paré[1,2,7,8]*

[1]Population Health Research Institute, David Braley Cardiac, Vascular and Stroke Research Institute, Hamilton, Canada; [2]Thrombosis and Atherosclerosis Research Institute, David Braley Cardiac, Vascular and Stroke Research Institute, Hamilton, Canada; [3]Department of Medical Sciences, McMaster University, Hamilton, Canada; [4]Department of Biochemistry and Biomedical Sciences, McMaster University, Hamilton, Canada; [5]Department of Medicine, McMaster University, Hamilton Health Sciences, Hamilton, Canada; [6]Department of Kinesiology, McMaster University, Hamilton, Canada; [7]Department of Pathology and Molecular Medicine, McMaster University, Michael G. DeGroote School of Medicine, Hamilton, Canada; [8]Department of Health Research Methods, Evidence, and Impact, McMaster University, Hamilton, Canada

*For correspondence: pareg@mcmaster.ca

**Abstract** Testosterone products are prescribed to males for a variety of possible health benefits, but causal effects are unclear. Evidence from randomized trials are difficult to obtain, particularly regarding effects on long-term or rare outcomes. Mendelian randomization analyses were performed to infer phenome-wide effects of free testosterone on 461 outcomes in 161,268 males from the UK Biobank study. Lifelong increased free testosterone had beneficial effects on increased bone mineral density, and decreased body fat; adverse effects on decreased HDL, and increased risks of prostate cancer, androgenic alopecia, spinal stenosis, and hypertension; and context-dependent effects on increased hematocrit and decreased C-reactive protein. No benefit was observed for type 2 diabetes, cardiovascular or cognitive outcomes. Mendelian randomization suggests benefits of long-term increased testosterone should be considered against adverse effects, notably increased prostate cancer and hypertension. Well-powered randomized trials are needed to conclusively address risks and benefits of testosterone treatment on these outcomes.

## Introduction

In developed countries, rising rates of both serum testosterone level testing and therapy initiation have been observed among older male patients (*Handelsman, 2013*; *Layton et al., 2014*). In the USA alone, it is estimated 1.5–1.7% of males are prescribed testosterone (*Baillargeon et al., 2018*; *Jasuja et al., 2017*). Randomized clinical trials (RCT) have attempted to elucidate the benefits and risks of testosterone treatment (*Bhasin et al., 2018a*; *Gagliano-Jucá and Basaria, 2019*). These studies identified short-term beneficial effects on bone mineral density (BMD), sexual function, body fat and muscle mass, and anaemia; potential adverse effects on venous thrombosis and coronary artery plaque; and no effects on cognitive function, fatigue, or hemoglobin A1$_c$ (HbA1$_c$) (*Bhasin et al., 2018a*; *Gagliano-Jucá and Basaria, 2019*; *Mohler et al., 2018*; *Snyder et al., 2018*). However, given the logistic and financial challenges involved in a well-powered RCT with appropriate follow-up, there is unlikely to be satisfactory evidence regarding long-term effects and risks of

**eLife digest** Men experience a gradual decline in their testosterone levels as they grow older. However, the effects of testosterone and the consequences of supplementation on the human body have been unclear.

Scientists use so-called randomized controlled trials to establish cause-and-effect and to reduce bias. In these experiments, participants are randomly assigned to a either a treatment group (that receives the intervention being tested) or a control group (that either receives an alternative intervention, a dummy or placebo, or no intervention at all).

Randomization ensures that both groups are balanced, and any resulting differences can be attributed to the treatment. However, randomized controlled trials are time-consuming and expensive, so trials of testosterone have had relatively small numbers of participants and short follow-up periods. This makes it difficult to draw conclusions about any potential effects of testosterone administration on less common diseases in men.

Now, Paré et al. investigated the effects of naturally produced testosterone using Mendelian randomization, which mimics randomized trials by exploiting the fact that parents randomly pass on their unique genetic variants to their children at conception. This random assignment of genetic variants leads to its informal namesake, "nature's clinical trial", and provides the ability to study cause-and-effect for any genetically determined factors, such as testosterone levels.

Paré et al. studied the long-term effects of testosterone on 22 diseases previously explored in randomized controlled trials, and hundreds of other traits and diseases that have not been investigated in any randomized controlled trials yet.

The Mendelian randomization analysis made it possible to examine the effects of lifelong naturally elevated testosterone levels on 469 traits and diseases. Paré et al. found that testosterone increased the density of bone mineral and decreased body fat. However, it also increased the risks of prostate cancer, high blood pressure, baldness and a condition affecting the spine. It also increased the number of red blood cells and decreased a marker of inflammation, which may be beneficial or detrimental depending on the context.

This shows that genetic analyses can be powerful methods to prioritize the allocation of limited resources towards investigating the most pressing clinical questions. The results of this study may help inform physicians and patients about the effects of long-term testosterone use. Ultimately, large randomized controlled trials are needed to conclusively address the cause-and-effect on these diseases.

adverse outcomes, such as myocardial infarction (MI), stroke and cancer (*Gagliano-Jucá and Basaria, 2019*). Given the rates of testosterone prescription, efforts to resolve the causal effects of testosterone on health outcomes have important public health implications (*Bhasin et al., 2018a*).

Mendelian randomization (MR) is a technique for causal inference that leverages the random allocation of genetic variants to infer the unconfounded relationship between an exposure and outcome. Similar to the random assignment of participants to experimental groups in a RCT, genetic variants are randomly allocated at meiosis (*Davies et al., 2018*). For instance, if individuals genetically randomized to produce higher testosterone develop different rates of cardiovascular disease (CVD), then MR analysis supports a causal effect of testosterone on risk of CVD (*Figure 1—figure supplement 1*). Notably, this technique has previously replicated RCT findings, among others demonstrating causal roles for LDL cholesterol and dysglycemia on CVD risk (*Holmes et al., 2015*; *Ross et al., 2015*). Earlier MR studies investigating the effects of testosterone have demonstrated harmful effects on lipid levels but inconsistent effects on CVD, and they were limited by the small number of genetic variants (*Schooling et al., 2018*; *Zhao et al., 2014*). A recent MR study using the UK Biobank identified a large number of genetic variants associated with testosterone and found evidence for harmful effects on several types of cancers but sex-specific effects on type 2 diabetes (T2D) (*Ruth et al., 2020*). This study highlighted the importance of performing sex-specific analyses for testosterone, but it was focused on glycemic and oncologic traits (*Ruth et al., 2020*). Therefore, we sought to expand the scope of prior studies by performing a comprehensive scan of the effects of free testosterone on human disease in males.

We hypothesized that MR and genetic risk score (GRS) analyses would enable estimation of the causal effects of longstanding exposure to high levels of free testosterone on health outcomes in males. We first conducted a genome-wide association study (GWAS) for calculated free testosterone (CFT) in male participants of the UK Biobank (n = 161,268) cohort to identify genetic determinants of free testosterone levels. Then, using MR, we investigated the causal effects of lifelong genetically-elevated free testosterone levels on a priori health outcomes previously investigated in RCTs of testosterone treatment, encompassing: expected clinical benefits (physical activity, strength, fat-free body mass, body fat, BMD, dementia, depression) and potential adverse effects (androgenic alopecia, heematocrit, T2D, prostate cancer, benign prostate hyperplasia, blood pressure, CVD, heart failure, ischemic stroke) (*Figure 1*; *Bhasin et al., 2018a*; *Gagliano-Jucá and Basaria, 2019*; *Mohler et al., 2018*; *Snyder et al., 2018*). Finally, we used GRS to investigate the associations of lifelong genetically-elevated free testosterone levels on 439 health outcomes, encompassing diseases (n = 415) and biomarkers of health (n = 24) (*Figure 1*).

## Results

### Genetic determinants of CFT in males

To calculate free testosterone levels, 187,524 males in the white, British subset of the UK Biobank cohort were excluded if they had missing levels of total testosterone, SHBG and albumin, or self-reported taking androgen medications. After these exclusions, the study population consisted of 161,268 males with an average CFT of 0.210 nmol/L (*Supplementary file 1* - Table 1 and *Figure 1—figure supplement 2*).

There were 13,338 genetic variants associated with CFT that reached genome-wide significance ($p<5\times10^{-8}$). After removing genetic variants associated with natural-log-transformed SHBG, there were 7048 genetic variants that comprised 93 independent signals carried forward for subsequent genetic analyses (*Supplementary file 1* - Table 2 and *Figure 1—figure supplement 3*). Overall, chip-based heritability of CFT was estimated at 15% (95% CI = 14 to 16), while these 93 independent genetic variants associated with CFT explained 3.7% of the total variance of CFT levels in males from the UK Biobank.

### Effect of genetically-predicted free testosterone on 22 a priori health outcomes

In males from the UK Biobank, sample size for the quantitative risk factors ranged from 30,439 to 156,403, while number of cases for dichotomous outcomes ranged from 1003 to 70,283 (*Table 1*). After adjusting for the 22 outcomes tested, one-sample MR analysis using IVW regression identified significant effects of CFT on hematocrit percentage, body fat-free percentage, body fat percentage, heel BMD, androgenic alopecia, and prostate cancer (*Table 1*). Each 0.1 nmol/L higher CFT had beneficial effects on increased heel BMD (0.40 SD; 95% CI = 0.25 to 0.54; $p=1.10\times10^{-7}$), increased body fat-free percentage (1.91%; 95% CI = 1.48 to 2.35; $p=9.06\times10^{-18}$), and decreased body fat percentage ($-1.88\%$; 95% CI = $-2.31$ to $-1.45$; $p=1.65\times10^{-17}$), but deleterious effects on increased hematocrit percentage (1.37%; 95% CI = 1.12 to 1.62; $p=1.03\times10^{-27}$), risk of prostate cancer (OR = 1.51; 95% CI = 1.21 to 1.88; $p=2.1\times10^{-4}$), and risk of androgenic alopecia (OR = 1.49; 95% CI = 1.19 to 1.86; $p=5.28\times10^{-4}$) (*Figure 3—figure supplements 1–6*). Leave-one-out analyses did not identify any outlying individual genetic variants responsible for the observed effects on any significant outcomes.

Sensitivity analyses were performed to detect violations of MR assumptions. Egger regression did not detect evidence of directional pleiotropy for any outcomes ($p_{intercept} <0.05$) (*Supplementary file 1* - Table 3). Results using MR-RAPS were consistent with IVW regression method for all significant outcomes (*Supplementary file 1* – Table 4). However, MR-PRESSO detected evidence of pleiotropic variants for hematocrit percentage, body fat-free percentage, body fat percentage, heel BMD, androgenic alopecia, whole body fat-free mass, hemoglobin A1C, glucose, handgrip strength, systolic blood pressure, diastolic blood pressure, T2D, and benign prostate hyperplasia (*Supplementary file 1* - Table 5). However, removal of pleiotropic variants made no changes to the significance or interpretation of earlier results using IVW regression (*Supplementary file 1* - Table 5).

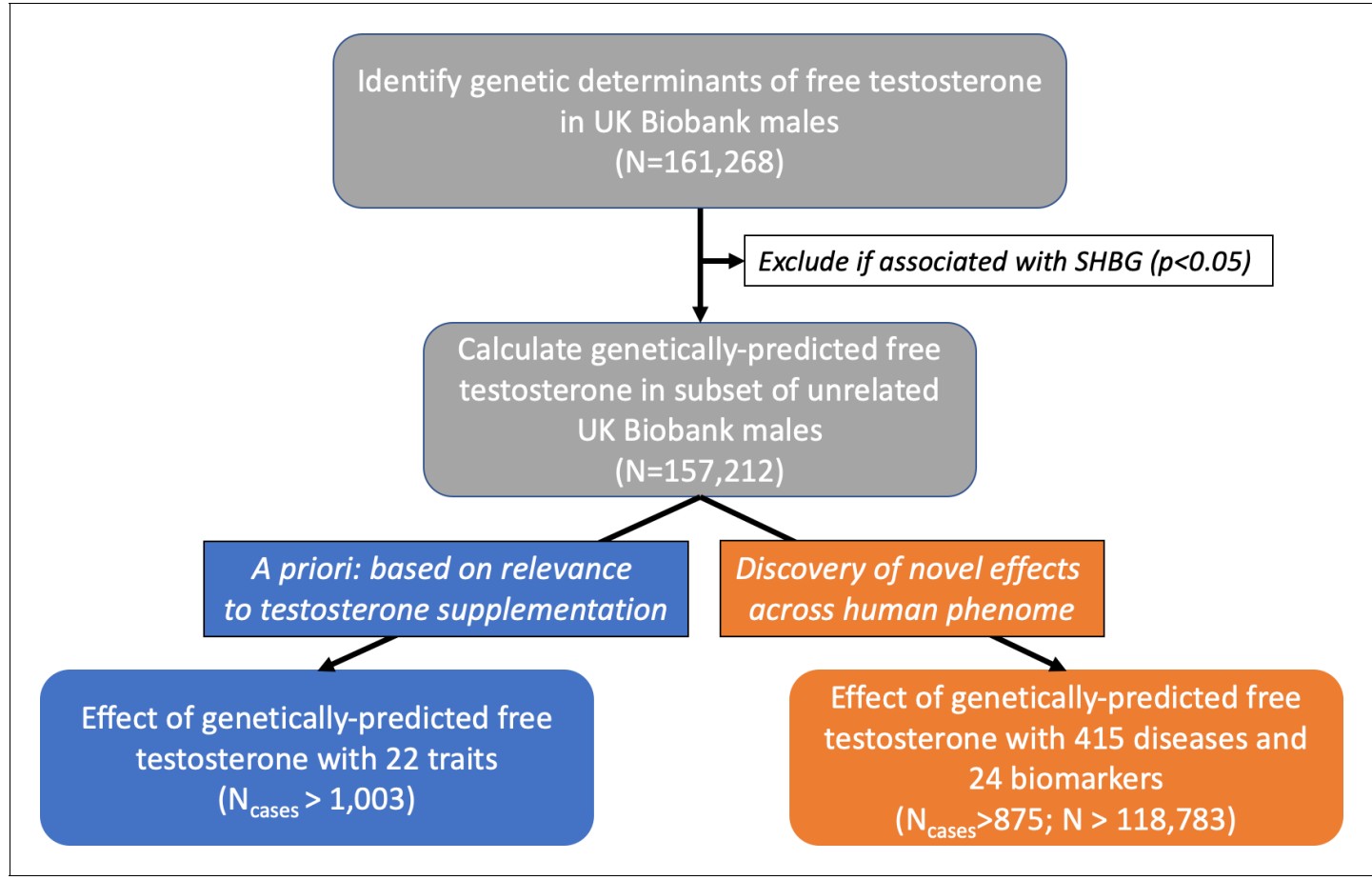

**Figure 1.** Flowchart depicting overall study design. Free testosterone levels were calculated in males from the UK Biobank cohort. Then, genetic variants were tested for association with levels of CFT and carried forward if: genome-wide significant ($p<5\times10^{-8}$) and unassociated with SHBG ($p<0.05$). In the subset of unrelated males, these genetic variants were used to investigate the effect of genetically-predicted CFT on: (1) 22 a priori outcomes relevant to suspected effects of testosterone treatment using Mendelian randomization, and (2) 439 outcomes in a hypothesis-free approach using a weighted genetic risk score. CFT, calculated free testosterone; MR, Mendelian randomization; SHBG, sex hormone-binding globulin.

The online version of this article includes the following figure supplement(s) for figure 1:

**Figure supplement 1.** Comparison of randomized controlled trial (RCT) and Mendelian randomization (MR) study designs demonstrating the common foundation behind interpretation of a causal effect of testosterone on cardiovascular disease (CVD).

**Figure supplement 2.** Distribution of free testosterone levels calculated using the Vermeulen equation in males from the UK Biobank cohort.

**Figure supplement 3.** Manhattan plot showing distribution of p-values from genome-wide association study of calculated free testosterone after exclusion of SHBG-associated variants based on chromosomal location.

**Figure supplement 4.** Distribution of sex hormone-binding globulin in males from the UK Biobank.

**Figure supplement 5.** Quantile-quantile plot for genome-wide association study of calculated free testosterone levels (before exclusion of SHBG-associated genetic variants).

**Figure supplement 6.** Distribution of total testosterone levels in males from the UK Biobank cohort.

## Phenome-wide effects of genetically-predicted free testosterone

To discover novel effects of free testosterone, we tested for the association of a GRS for testosterone with 415 diseases and 24 biomarkers in the same subpopulation of unrelated males from the UK Biobank. Sample size for biomarkers ranged from 118,783 for lipoprotein(a) to 149,940 for total cholesterol, while number of cases for diseases ranged from 876 for 'localized superficial swelling, mass, or lump' to 40,960 for 'hypertension' (*Figure 2—source data 1*). After adjusting for the 439 outcomes tested, each 0.1 nmol/L increase in genetically-predicted CFT was significantly associated with beneficial effects on lowered C-reactive protein (β = −0.085 SD; 95% CI = −0.119 to −0.052; p=$6.15\times10^{-7}$) but adverse effects on increased creatinine (β = 0.113 SD; 95% CI = 0.079 to 0.146; p=$4.78\times10^{-11}$), lowered apolipoprotein A (β = −0.018 g/L; 95% CI = −0.026 to −0.01;

**Table 1.** Effect of calculated free testosterone on 22 health outcomes from the UK Biobank relevant to effects of testosterone treatment in males.

| Outcome | Effect per 0.1 nmol/L increased CFT (95% CI) | P-value | Sample Size Cases/Controls |
|---|---|---|---|
| **Outcomes with Expected Clinical Benefits** | | | |
| Body fat-free percentage* | 1.91% (1.48 to 2.35) | 9.06E-18 | 154254 |
| Body fat percentage* | −1.88% (−2.31 to −1.45) | 1.65E-17 | 153772 |
| Heel bone mineral density* | 0.40 SD (0.25 to 0.54) | 1.10E-07 | 90676 |
| Depression | OR = 1.45 (1.1 to 1.91) | 7.77E-03 | 4725/152485 |
| Accelerometer-based physical activity | 0.89 milligravity (−0.05 to 1.82) | 0.06 | 30439 |
| All fracture | OR = 0.89 (0.71 to 1.11) | 0.30 | 9133/148077 |
| Handgrip strength | 0.29 kg (−0.31 to 0.89) | 0.34 | 156400 |
| All dementia | OR = 1.26 (0.67 to 2.34) | 0.47 | 1003/156207 |
| **Outcomes with Potential Adverse Effects** | | | |
| Hematocrit percentage* | 1.37% (1.12 to 1.62) | 1.03E-27 | 152872 |
| Prostate cancer* | OR = 1.51 (1.21 to 1.88) | 2.10E-04 | 7586/149624 |
| Androgenic alopecia* | OR = 1.49 (1.19 to 1.86) | 5.28E-04 | 70283/85756 |
| Benign prostatic hyperplasia | OR = 1.36 (1.10 to 1.67) | 3.80E-03 | 10894/146316 |
| Myocardial infarction | OR = 1.23 (1 to 1.53) | 0.05 | 9398/147812 |
| Glucose | −0.06 mmol/L (−0.14 to 0.02) | 0.12 | 138307 |
| Hemoglobin A1c | −0.34 mmol/mol (−0.82 to 0.15) | 0.17 | 149828 |
| All stroke | OR = 1.18 (0.90 to 1.56) | 0.23 | 4569/152641 |
| Diastolic blood pressure | 0.27 mmHg (−0.30 to 0.85) | 0.35 | 148384 |
| Ischemic stroke | OR = 0.92 (0.61 to 1.37) | 0.67 | 2122/155088 |
| Systolic blood pressure | −0.12 mmHg (−1.23 to 1.00) | 0.84 | 148383 |
| Type 2 diabetes | OR = 1.02 (0.81 to 1.28) | 0.87 | 11079/146131 |
| Venous thromboembolism | OR = 1.02 (0.74 to 1.4) | 0.92 | 4127/153083 |
| Heart failure | OR = 1.01 (0.76 to 1.34) | 0.95 | 4288/152922 |

* Significant adjusting for Bonferroni correction of 22 outcomes ($p<2.27\times10^{-3}$).

CFT, calculated free testosterone.

p=$1.55\times10^{-5}$), lowered HDL (β = −0.074 SD; 95% CI = −0.109 to −0.039; p=$3.62\times10^{-5}$), and increased risks of hypertension (OR = 1.17; 95% CI = 1.08 to 1.26; p=$2.83\times10^{-5}$), and spinal stenosis (OR = 2.03; 95% CI = 1.51 to 2.75; p=$3.82\times10^{-6}$) (*Table 2* and *Figure 2*).

As confirmation, we demonstrated the GRS was indeed not associated with natural log-transformed natural log-transformed SHBG levels in males (p=0.12). For all statistically significant outcomes, associations were directionally consistent after removing participants taking blood pressure medication (*Supplementary file 1* - Table 6) or cholesterol-lowering medication (*Supplementary file 1* - Table 7). Further sensitivity analyses were performed by repeating the one-sample MR analysis using 52 genetic variants associated with total testosterone in males from the UK Biobank (*Supplementary file 1* - Table 8). For all statistically significant outcomes, effects observed using total testosterone genetic variants were directionally consistent with CFT, and results for all outcomes are presented in *Supplementary file 1* - Tables 9 and 10. Finally, most effect estimates for genetically-predicted testosterone in this stu dy were comparable in magnitude to effect sizes reported in RCTs except bone mineral density (*Figure 3*).

## Discussion

We herein perform MR and GRS analyses of CFT to identify effects of endogenous free testosterone in males on 461 health outcomes. All effects are reported in terms of 0.1 nmol/L of CFT to

**Table 2.** Effects of calculated free testosterone on 439 health outcomes in males from the UK Biobank significant after adjusting for multiple hypothesis testing using Bonferroni correction (p<1.14×10$^{-4}$).

| Outcome | Effect per 0.1 nmol/L increased CFT (95% CI) | P-value | Sample Size Cases/Controls |
|---|---|---|---|
| Creatinine | 0.113 SD (0.079 to 0.146) | $4.78 \times 10^{-11}$ | 149849 |
| C-reactive protein | −0.085 SD (−0.119 to −0.052) | $6.15 \times 10^{-7}$ | 149547 |
| Spinal stenosis | OR = 2.03 (1.51 to 2.75) | $3.82 \times 10^{-6}$ | 1917/150919 |
| Apolipoprotein A | −0.018 g/L (−0.026 to −0.01) | $1.55 \times 10^{-5}$ | 138185 |
| HDL cholesterol | −0.074 SD (−0.109 to −0.039) | $3.62 \times 10^{-5}$ | 138394 |
| Essential hypertension | OR = 1.17 (1.08 to 1.27) | $7.53 \times 10^{-5}$ | 40809/115957 |
| Hypertension | OR = 1.17 (1.08 to 1.26) | $1.05 \times 10^{-4}$ | 40960/115957 |

CFT, calculated free testosterone; HDL, high density lipoprotein; GRS, genetic risk score.

approximate expected effect sizes after initiation of testosterone treatment (*Bhasin et al., 2018b*). Among 22 a priori outcomes with suspected effects based on RCTs of testosterone treatment, MR analyses demonstrated that each 0.1 nmol/L increase in CFT was associated with adverse effects on

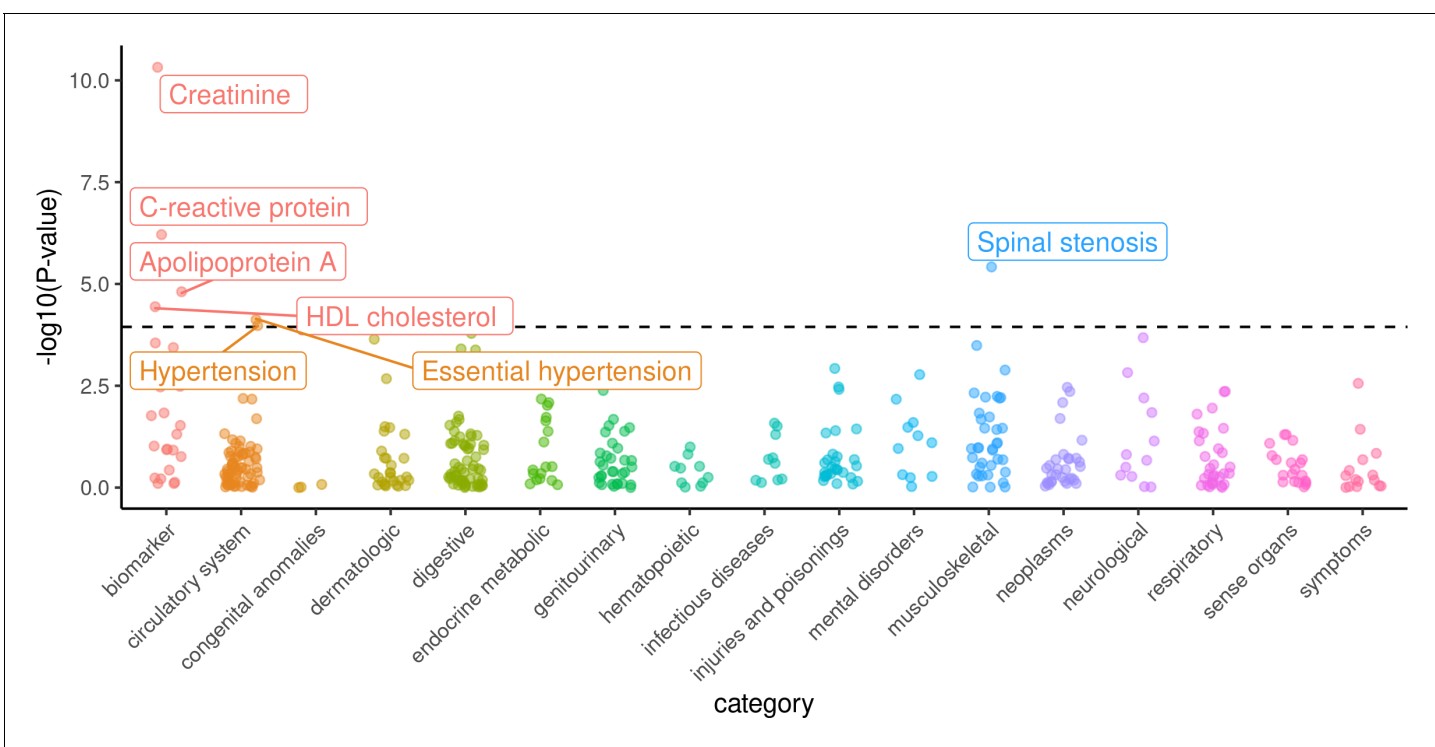

**Figure 2.** Phenome-wide survey of effects of genetically-predicted calculated free testosterone on 439 health outcomes in males from the UK Biobank. Logistic or linear regression was used to assess the association of the genetic score for free testosterone against each dichotomous or quantitative outcome, respectively. -log$_{10}$(p-values) for the association of each outcome on the y-axis are stratified into subcategories on the x-axis. Labelled outcomes were statistically significant adjusting for multiple hypothesis testing (p<1.14×10$^{-4}$).
The online version of this article includes the following source data for figure 2:

**Source data 1.** Associations of genetically-predicted calculated free testosterone for 439 health outcomes across the human phenome.

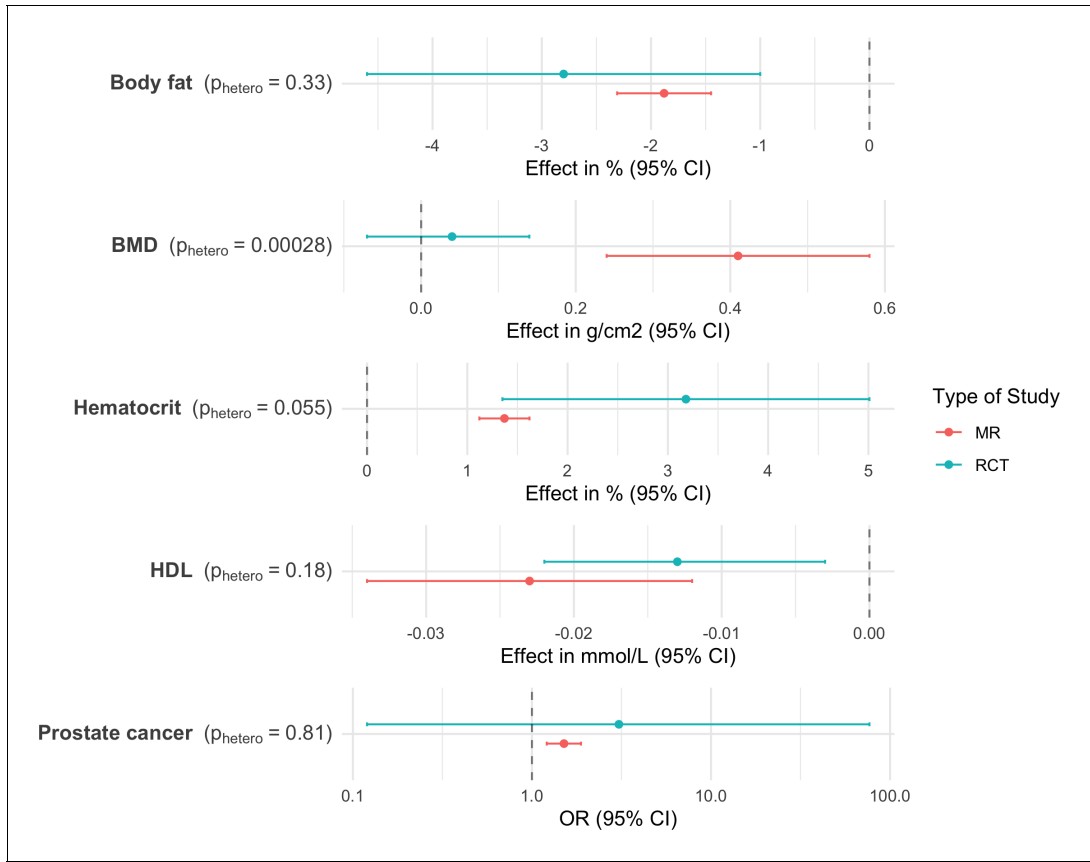

**Figure 3.** Comparison of effect sizes reported in randomized controlled trials and Mendelian randomization analyses. Error bars indicate 95% confidence intervals around the effect estimate. MR effect estimates are reported in terms of 0.1 nmol/L of CFT to approximate expected effect sizes after initiation of testosterone treatment (*Bhasin et al., 2018b*).

The online version of this article includes the following figure supplement(s) for figure 3:

**Figure supplement 1.** Comparison of effect of calculated free testosterone on hematocrit percentage using Mendelian randomization with IVW and Egger regression methods.

**Figure supplement 2.** Comparison of effect of calculated free testosterone on body fat-free percentage using Mendelian randomization with IVW and Egger regression methods.

**Figure supplement 3.** Comparison of effect of calculated free testosterone on body fat percentage using Mendelian randomization with IVW and Egger regression methods.

**Figure supplement 4.** Comparison of effect of calculated free testosterone on heel bone mineral density using Mendelian randomization with IVW and Egger regression methods.

**Figure supplement 5.** Comparison of effect of calculated free testosterone on prostate cancer using Mendelian randomization with IVW and Egger regression methods.

**Figure supplement 6.** Comparison of effect of calculated free testosterone on androgenic alopecia using Mendelian randomization with IVW and Egger regression methods.

increased risk of prostate cancer, risk of androgenic alopecia, and hematocrit percentage, but beneficial effects on increased heel BMD, increased body fat-free percentage and decreased body fat percentage. Findings on body composition, hematocrit, and BMD are consistent with short-term effects in randomized trials of testosterone treatment (*Bhasin et al., 2018a*). Although testosterone treatment has not been conclusively shown to increase risk of prostate cancer and androgenic alopecia in RCTs, androgen suppression therapies, such as of 5α-reductase inhibitors, are used as treatment for androgenic alopecia and prostate cancer (*Adil and Godwin, 2017*; *Andriole et al., 2010*). The increased risk of prostate cancer replicates effects of testosterone observed in a previous MR analysis using independent data from the PRACTICAL consortium, and further supports the role of testosterone in development of these outcomes. As the leading cause of cancer among men, the predicted 1.5-fold increased risk as a result of changes in testosterone observed after initiation of

testosterone treatment warrants further investigation in clinical trials and greater scrutiny in at-risk patient populations (*American Cancer Society, 2019*; *Bhasin et al., 2018b*). Furthermore, these results cast doubt on cardiovascular, cognitive, or metabolic benefit for increased testosterone, as we do not find evidence of a beneficial effect of CFT on hard endpoints, such as dementia, MI, stroke, fractures, or T2D (*Aukrust et al., 2009*). Most of the estimates from MR analyses were comparable with effect sizes from RCTs (*Figure 3*). There was only significant heterogeneity between the effects on BMD for MR and RCT, but it is difficult to make direct comparisons due to variable change in testosterone levels after administration of testosterone in each RCT, different methods and anatomical sites of BMD estimation, and differences between short-term effects in RCTs relative to life-long effects in MR.

Among the remaining outcomes without well-established effects from RCTs, we identified evidence of novel associations between an increased GRS for CFT with adverse effects on creatinine, HDL, apolipoprotein A, hypertension, and spinal stenosis, but beneficial effects on C-reactive protein. Higher genetically-predicted free testosterone was associated with increased creatinine ($\beta$ = 0.113 SD; 95% CI = 0.079 to 0.146; p=4.78$\times10^{-11}$). Mechanistically, effects of testosterone on renal function are unclear, but this effect may be mediated through the known effect of testosterone on increased muscle mass which is tightly related to serum creatinine (*Carrero et al., 2009*; *Filler et al., 2016*; *Schutte et al., 1981*). HDL cholesterol ($\beta$ = −0.074 SD; 95% CI = −0.109 to −0.039; p=3.62$\times10^{-5}$) and its main protein component, apolipoprotein A ($\beta$ = −0.018 g/L; 95% CI = −0.026 to −0.01; p=1.55$\times10^{-5}$), were both decreased with higher genetically-predicted free testosterone. Likewise, the Testosterone Trials found male participants over 65 years of age randomized to testosterone experienced mildly lowered HDL cholesterol levels after 12 months (*Mohler et al., 2018*; *Snyder et al., 2018*). Higher free testosterone was associated with decreased C-reactive protein (CRP) ($\beta$ = −0.085 SD; 95% CI = −0.119 to −0.052; p=6.15$\times10^{-7}$). Although the Testosterone Trials did not find any change in CRP in its testosterone arm, testosterone is widely-believed to have suppressive effects on the immune system which may extend to markers of inflammation such as CRP (*Trigunaite et al., 2015*). Furthermore, despite no effect on SBP or DBP, our analyses suggest 0.1 mol/L higher free testosterone is associated with increased risk of hypertension (OR = 1.17; 95% CI = 1.08 to 1.27; p=1.05$\times10^{-4}$). Given the multifactorial nature of this disease, the apparent discrepancy between blood pressure and hypertension may be explained by an effect on other risk factors that develop into hypertension. Moreover, both human and animal studies suggest a role of testosterone on hypertension. A randomized controlled trial found testosterone administration increased levels of NT-proBNP, and studies of both transgender men and anabolic steroid users have found testosterone increased arterial stiffness and blood pressure (*Bachmann et al., 2019*; *Hartgens and Kuipers, 2004*; *Velho et al., 2017*). Meanwhile, animal models have shown testosterone may aggravate hypertension and exacerbate increased production of reactive oxygen species specifically in hypertensive but not normotensive rat vascular endothelial tissue (*Chignalia et al., 2012*; *Reckelhoff et al., 1998*). Testosterone is widely-believed to have anti-inflammatory and osteogenic effects, but our analyses showed an association with higher risk of spinal stenosis (OR = 2.03; 95% CI = 1.51 to 2.75; p=3.82$\times10^{-6}$). However, the literature shows some evidence that higher testosterone is associated with greater loss of cartilage in healthy older males, and evidence from mouse models suggest testosterone has a sex-specific role in worsening osteoarthritis, a common risk factor for spinal stenosis (*Hanna, 2005*; *Hl et al., 2007*).

In comparison to previous MR studies, our results broaden the scope of the existing literature by comprehensively assessing the effects of testosterone on 461 health outcomes including hard endpoints and intermediate biomarkers. Moreover, a key strength of this study was the stringent attempt to control for pleiotropic effects of SHBG on free testosterone by conservatively removing any genetic variants in the GRS that were associated with SHBG (p<0.05). The apparent difference between protective effects of testosterone observed in a previous MR analysis of testosterone and lack of protective effect in our study might be a result of less stringent control for pleiotropic effects of SHBG in the previous study. Given studies have identified associations between SHBG and risk of T2D independent of testosterone and a direct role of SHBG in mediating signalling on target cells, insufficient controls for SHBG may lead to residual pleiotropic effects (*Lakshman et al., 2010*; *Rosner et al., 2010*; *Vikan et al., 2010*). Other reasons may include genetic variants explaining less variation in testosterone levels in our study, fewer cases of T2D leading to inadequate statistical

power to detect weaker effects in our study, or other differences between the populations of the UK Biobank in our study and DIAGRAM consortium used by *Ruth et al., 2020*.

There are several limitations of this study. First, an assumption of the MR analysis is that the effect of the genetic variant on the outcome occurs only through free testosterone levels, such that there are no pleiotropic effects through other proteins or mechanisms (*Davies et al., 2018*). This concern was minimized by the use of multiple genetic variants, which limited the likelihood of a common alternative pathway confounding our observation. Moreover, we performed several sensitivity analyses and excluded genetic variants associated with SHBG levels, which is a potential source of pleiotropy through its effects on other hormones. Although a stringent p-value threshold was selected for genetic variants, the winner's curse phenomenon may still bias genetic effect sizes due to the same sample being used to select genetic variants and estimate effect sizes on testosterone. Additionally, one-sample MR may be susceptible to bias towards the confounded estimate if the genetic variants are 'weak instruments', which can occur if the genetic variants don't explain enough of the variance in free testosterone levels (*Davies et al., 2018*). To address this concern, we confirmed the selected genetic variants were strong instruments using a common threshold in MR literature (F-statistic >10) (*Davies et al., 2018*). Next, the UK Biobank is generally healthier and higher socioeconomic status than the general population, so there are insufficient cases to detect effects on certain rarer outcomes, such as Alzheimer's disease, and inadequate power to identify weaker effects of free testosterone on common outcomes. Relatedly, an inherent limitation for outcomes ascertained using linked electronic medical records is a lack of adjudication and consistent application of codes in clinical practice. In the UK Biobank, CFT levels were below the reference ranges for young healthy individuals, which may be attributable to the older age of the cohort and inherent inaccuracy of immunoassays at lower levels of total testosterone. Total testosterone levels are similarly low relative to reference ranges and comparable to previous studies in the UK Biobank (*Peila et al., 2020*; *Petermann-Rocha et al., 2020*). Additional sources of variability introduced into the total testosterone measurements include differences in fasting times, diets, and time of day at which blood was drawn from participants. Nevertheless, genetic variants associated with testosterone consistently replicated known effects of testosterone on established outcomes, such as body fat, body fat-free mass, and hematocrit (*Table 1*). Furthermore, although the free hormone hypothesis is still debated by experts, we found largely consistent effects on outcomes using genetically-predicted free testosterone and total testosterone (*Handelsman, 2017*). The only significant outcomes from MR analyses with free testosterone that showed no significant effect with total testosterone across all MR methods were HDL (p=0.55) and apolipoprotein A (p=0.45). Finally, these results represent lifelong effects of endogenous free testosterone and may not necessarily reflect effects of exogenous testosterone treatment, which can vary in duration, age of initiation, and dosage.

Taken altogether, the decision to initiate long-term testosterone use warrants careful consideration of benefits and risk. Beneficial effects on body composition, sexual function, hematocrit, and BMD should be weighed against detrimental effects on androgenic alopecia, prostate cancer, hypertension and spinal stenosis, and no detectable beneficial effects on other major clinical endpoints. Ultimately, well-designed and appropriately powered RCTs, such as the ongoing TRAVERSE trials (clinicaltrials.gov, NCT03518034), are necessary to conclusively address questions of safety and effectiveness of testosterone treatment. However, as demonstrated in this study, genetically-informed analyses can be powerful tools to aid health professionals in prioritizing allocation of limited resources towards investigating the most pressing questions.

## Materials and methods

### Study population - UK Biobank

The UK Biobank is a large-scale longitudinal cohort study that recruited over 500,000 people between the ages of 37–73 across the United Kingdom from 2006 to 2010 (*Sudlow et al., 2015*) (RRID:SCR_012815). UK Biobank received ethical approval from the North West Multi-Centre Research Ethics Committee (REC reference: 11/NW/0382). This research was conducted using the UK Biobank under Application Number 15255. For this study, UK Biobank participants were included if white British ancestry, and no self-reported androgen medication at recruitment based on field ID 20003.

## Measurement of testosterone and sex hormone-binding globulin in UK Biobank

In the UK Biobank, total testosterone and sex hormone-binding globulin (SHBG) were measured on a Beckman Coulter Unicel DXI 800 using a one-step competitive analysis and two-step sandwich immunoassay, respectively. Analytical range for the immunoassays of total testosterone and SHBG were 0.35 to 55.52 and 0.33 to (226-242) nmol/L, respectively. For total testosterone, within-laboratory CV for high, medium, and low concentration quality control samples were 4.15, 3.66, and 8.34%. For SHBG, within-laboratory CV for high, medium, and low concentration quality control samples were 5.22, 5.25, and 5.67%. For each blood sample drawn at recruitment, testosterone, SHBG, and albumin were each measured only once. Testosterone and SHBG measurements were flagged if they fell outside the manufacturer's observed reportable range, or samples reported high levels of bilirubin, hemoglobin or lipids/turbidity that might interfere with the assay. Testosterone measurements were flagged if levels of total protein ($<55$ or$>85$ g/L) or triglycerides ($>20$ mmol/L) could interfere with the assay measurements. To monitor assay consistency, all samples were run with internal quality control samples between batches and operations used external quality assurance schemes against the ISO 17025:2005 standard.

## Genome-wide association study of CFT

Individual-level genetic data was available for 488,317 participants that consented to blood collection and genotyping. Genotyping was performed with the Applied Biosystems UK Biobank Lung Exome Variant Evaluation (UK BiLEVE) and UK Biobank Axiom arrays (Affymetrix Research Services Laboratory, Santa Clara, California, USA). Description of quality control has been previously described in detail (*Bycroft et al., 2017*). Genetic variants located in the human leukocyte antigen gene complex were excluded due to extensive pleiotropic effects.

For genome-wide association testing, samples were restricted to a subset of 161,268 males with white British ancestry, no androgen medication (n = 2,137), and no missing values of testosterone, SHBG, or albumin at recruitment. Free testosterone at recruitment was calculated using the Vermeulen equation (*Vermeulen et al., 1999*). CFT levels were winsorized such that outlying values greater or less than four standard deviations (SD) away from the mean in males were set to 4 SD.

This study was restricted to genetic variants from 'v3' release of the UK Biobank data including those present in the Haplotype Reference Consortium and 1000 Genomes panels with imputation imputation quality greater than 0.7, no deviation from Hardy-Weinberg equilibrium ($p>1\times10^{-10}$) and minor allele frequency greater than 1% (*McCarthy et al., 2016*). To allow for genetic relatedness between participants, linear mixed models in BOLT-LMM were used to test for associations of genetic variants (*Loh et al., 2015*). The model was adjusted for age, age$^2$, chip type, assessment center, and the first 20 genetic principal components. Genetic variants near the *SHBG* gene may alter binding affinity for testosterone thereby violating assumptions of the Vermeulen equation, or risk having pleiotropic effects through binding of other sex hormones (*Ohlsson et al., 2011*). Therefore, any genetic variants associated with CFT reaching genome-wide significance ($p\leq5\times10^{-8}$) were excluded if associated with natural log-transformed SHBG levels at a stringent threshold ($p<0.05$) in the same subset of the UK Biobank (*Figure 1—figure supplement 4*). To arrive at an independent set of genetic variants, variants associated with CFT but not SHBG were pruned based on linkage disequilibrium (LD) at a threshold of r$^2$ <0.01 using Europeans from 1000 Genomes phase three as reference panel (*Abecasis et al., 2012*) (RRID:SCR_006828).

Genomic inflation factor (λ) was 1.2 and calculated as the ratio of the median test statistic from the GWAS relative to the expected median test statistic under a null model (*Figure 1—figure supplement 5*). To distinguish between an inflated λ due to population stratification or polygenic inheritance of the trait, the intercept of an LD score regression line was determined to be 1.03 indicating the observed inflation could be attributed to polygenicity rather than uncontrolled population stratification. LD score regression was performed and intercept was calculated with LDSC software (*Bulik-Sullivan et al., 2015*) using 1000 Genomes Europeans phase three data as the LD reference panel (*Abecasis et al., 2012*).

## Definition of health-related UK Biobank outcomes

For MR analyses, 22 health outcomes were selected a priori based on relevance with known or suspected effects of testosterone treatment and categorized based on expected beneficial or adverse effects from RCT data. Outcomes with expected beneficial effects were fractures at any site, heel BMD, body fat percentage, body fat-free percentage, dementia, depression, handgrip strength, and physical activity level measured by wrist-worn accelerometer. Outcomes with potential adverse effects were stroke, androgenic alopecia, benign prostate hyperplasia (BPH), blood pressure, glucose, hematocrit percentage, hemoglobin A1c, heart failure, prostate cancer, MI, type 2 diabetes (T2D), and venous thromboembolism. Depression was coded using a 'broad' definition as previously described, which included self-reported depressive symptoms with associated impairment, or having sought help for 'nerves, anxiety, tensions or depression' (*Howard et al., 2018*). Androgenic alopecia was defined based on participants' responses to the question, 'Which of the following best describes your hair/balding pattern?' (field ID 2395). Available options were four pictures of hair patterns (*Supplementary file 1 – Figure 1*). Individuals with pattern 3 or four were cases, pattern 1 and 2 were controls, and 'do not know' or 'prefer not to answer' responses were excluded. Physical activity was assessed using the overall acceleration average from wrist-worn accelerometer devices over the course of approximately 7 days. Following UK Biobank recommendations, individuals were excluded from the analysis based on poorly calibrated data (field ID: 90016) or having worn the device for insufficient time to get a stable measure of physical activity (field ID: 90015) (*Doherty et al., 2017*). Blood pressure measures were coded as the average of two automated measurements of blood pressure taken a few moments apart by a registered nurse using an Omron 705 IT electronic blood pressure monitor. Body fat percentage and whole body fat-free mass were estimated based on impedance measurements from a Tanita BC418MA body composition analyser. Heel BMD was estimated as a T-score based on quantitative ultrasound index through the calcaneus relative to that expected in someone of the same sex. Handgrip strength was calculated as the average of right and left hands measured using a Jamar J00105 hydraulic hand dynamometer. hemoglobin $A1_C$ was measured using high performance liquid chromatography analysis on a Bio-Rad VARIANT II Turbo. Glucose was measured using hexokinase analysis on a Beckman Coulter AU5800. Hematocrit percentage was measured using a Coulter LH750 and calculated as the relative volume of packed erythrocytes to whole blood, computed by the formula: $\frac{red\ blood\ cells\ *\ mean\ corpuscular\ volume}{10}$. Detailed descriptions of all 22 outcomes are shown in *Supplementary file 1* – Table 11.

For hypothesis-free GRS analyses, we included 24 blood biomarkers measured at recruitment and 415 diseases derived from linked electronic medical records (*Supplementary file 1* - Table 12; *Brion et al., 2013*; *Denny et al., 2013*; *Wu et al., 2019*). Disease outcomes were defined using the previously published 'PheCode' scheme to aggregate ICD-10 codes from hospital episodes (field ID 41270), death registry (field ID 40001 and 40002), and cancer registry (field ID 40006) records (*Denny et al., 2013*; *Wu et al., 2019*). Given the small number of cases for many disease outcomes, any outcomes with detectable odds ratios less than 0.5 or greater than 2 per 0.1 nmol/L at 80% power were excluded ($n_{cases}$ < 871) based on approximate changes in response to testosterone supplementation (*Bhasin et al., 2018b*; *Brion et al., 2013*; *Traustadóttir et al., 2018*). After these exclusions, there were 415 diseases that remained for subsequent analyses in this study. Furthermore, all blood biomarkers measured by the UK Biobank at recruitment were included except estradiol and rheumatoid factor, which were complicated by majority missing values below the limit of detection of the assay ($n_{biomarkers}$ = 24). Detailed descriptions of all 439 outcomes (415 diseases and 24 biomarkers) are shown in *Supplementary file 1* – Table 12.

## Mendelian randomization analysis

In a subset of unrelated males with White British ancestry, the association of all independent genetic variants associated with CFT were determined for each of the 22 a priori outcomes using additive genetic models in BGENIE v1.2 and adjusted for the same covariates as the model for CFT (*Bycroft et al., 2017*). For each of the 22 outcomes, one-sample MR analysis was used to combine the effect of each independent genetic variant on CFT with its effect on the outcome using the inverse variance-weighted (IVW) method (*Burgess et al., 2016*). Effect estimates were reported per 0.1 nmol/L increase in CFT levels based on approximate changes in response to testosterone treatment (*Bhasin et al., 2018b*). For dichotomous outcomes, odds ratios were approximated as

previously described (*Adams et al., 2018*) by converting linear effect estimates from BGENIE to log-odds scale using:

$$\log(OR) = \frac{}{k(1-k)},$$ where $k$ is the proportion of cases for the given outcome.

Given the polygenic nature of testosterone and potential for pleiotropy, for outcomes with statistically significant effects using the IVW method, standard sensitivity analyses were conducted to correct for pleiotropic effects, such as MR-Egger, MR-RAPS, and MR-PRESSO (*Bowden et al., 2015*; *Verbanck et al., 2018*). To investigate and correct for directional pleiotropy on each outcome, we performed Egger regression. For outcomes with y-intercept of the regression line significantly different from 0 (p<0.05), there was evidence of directional pleiotropy and the causal estimate from MR Egger was reported to attempt to control for pleiotropic effects (*Bowden et al., 2015*). As a sensitivity analysis robust to idiosyncratic pleiotropy and weak instrument bias, MR-RAPS (Robust Adjusted Profile Score) was conducted using overdispersion and Tukey's loss function (*Zhao et al., 2018*). To detect and correct for potential bias from invalid variants with pleiotropic effects, we performed the MR-PRESSO (Mendelian Randomization Pleiotropy RESidual Sum and Outlier) test with 10,000 simulations (*Verbanck et al., 2018*). The global test p-value evaluated whether there was any overall horizontal pleiotropy among all genetic variants. For outcomes with significant p-values (p<0.05), outlying genetic variants with predicted pleiotropic effects were removed and MR analysis repeated to correct for horizontal pleiotropy. The distortion test evaluated whether removal of the pleiotropic variants resulted in a significantly different causal estimate (p<0.05). Leave-one-out analysis was performed such that the IVW MR analysis was repeated after each genetic variant was excluded to identify effects on an outcome that are driven by a single outlying genetic variant. Furthermore, the set of genetic variants used in MR analysis were assessed for 'weak instrument bias', which can result in biased estimates if genetic variants don't explain enough variance in exposure (e. g., CFT) levels (*Pierce et al., 2011*). Lastly, as a sensitivity analysis, all MR and GRS analyses were repeated using genetic variants associated with total testosterone. Finally, for significant outcomes, we compared estimated effect sizes from this MR study with reported effect sizes from random controlled trials of testosterone therapy, where possible, in *Figure 3* (*Cui et al., 2014*; *Fernández-Balsells et al., 2010*; *Ng Tang Fui et al., 2016*; *Zhang et al., 2020*).

In consideration of 'weak instrument bias', the F-statistic was 66 for the genetic variants associated with CFT, which was considered a strong instrument based on the recommended threshold of greater than 10 (*Davies et al., 2018*). MR-PRESSO was performed using the *MR-PRESSO* package and all other MR analyses were implemented using the *TwoSampleMR* package (*Hemani et al., 2018*; *Verbanck et al., 2018*) (RRID:SCR_019010).

## Genetic risk score analysis

A genetically-predicted value of CFT was determined for each individual by constructing weighted GRS in the unrelated White British subset of UK Biobank males (n = 157,252). Weighted GRS were calculated by multiplying the effect of each CFT-associated genetic variant by the number of effect-corresponding alleles and summing this value for each individual. The GRS was tested for association with outcomes using logistic or linear regression models for case-control or quantitative outcomes, respectively, and adjusted for the same covariates as the GWAS for CFT. Effect estimates were reported per 0.1 nmol/L increase in CFT levels based on approximate changes in response to testosterone treatment (*Bhasin et al., 2018b*). As sensitivity analyses, we repeated GRS analyses after excluding males that self-reported taking blood pressure (n = 38,676) or cholesterol medication (n = 35,737) at recruitment based on field ID 6177.

## Genetic determinants and effects of total testosterone in males

As a set of sensitivity checks, we repeated all GWAS, MR, and GRS analyses using total testosterone. In the White British subset of the UK Biobank, there were 175,421 males with total testosterone measured with an average 11.9 nmol/L (*Figure 1—figure supplement 6*). In this population, a genome-wide association study was conducted for total testosterone as described herein for CFT. After removing genetic variants associated with natural-log-transformed SHBG and LD pruning for independent SNPs ($r^2$ <0.01), there were 52 independent genetic variants associated (p<$5\times10^{-8}$) with total testosterone in males from the UK Biobank (*Supplementary file 1* – Table 8).

All statistical analyses were performed under R version 3.6.0, unless otherwise specified (RRID: SCR_001905). A two-sided p-value less than $5 \times 10^{-8}$ for GWAS, $2.27 \times 10^{-3}$ (0.05/22 outcomes) for a priori MR analyses, and $1.14 \times 10^{-4}$ (0.05/439 outcomes) for hypothesis-free GRS analyses was considered statistically significant.

## Acknowledgements

The authors are thankful for all the participants that contributed to the UK Biobank study.

## Additional information

### Competing interests

Hertzel C Gerstein: HCG reports research grants from Eli Lilly, AstraZeneca, Merck, Novo Nordisk, and Sanofi; honoraria for speaking from AstraZeneca, Boehringer Ingelheim, Eli Lilly, Novo Nordisk, and Sanofi; and consulting fees from Abbott, AstraZeneca, Boehringer Ingelheim, Eli Lilly, Merck, Novo Nordisk, Janssen, Sanofi, Kowa, and Cirius. The other authors declare that no competing interests exist.

### Funding

| Funder | Grant reference number | Author |
|---|---|---|
| Canadian Institutes of Health Research | Frederick Banting and Charles Best Canada Graduate Scholarships Doctoral Award | Michael Chong |
| Canadian Institutes of Health Research | Post-Doctoral Fellowship | Robert W Morton |
| McMaster University | E.J. Moran Campbell Internal Career Research Award | Marie Pigeyre |
| McMaster University | McMaster-Sanofi Population Health Institute Chair in Diabetes Research and Care | Hertzel C Gerstein |
| Cisco Systems | Professorship in Integrated Health Biosystems | Guillaume Paré |
| Canada Research Chairs | Canada Research Chair in Genetic and Molecular Epidemiology | Guillaume Paré |

The funders had no role in study design, data collection and interpretation, or the decision to submit the work for publication.

### Author contributions

Pedrum Mohammadi-Shemirani, Data curation, Software, Formal analysis, Investigation, Visualization, Writing - original draft; Michael Chong, Data curation, Software, Formal analysis, Writing - review and editing; Marie Pigeyre, Hertzel C Gerstein, Conceptualization, Writing - review and editing, Analysis and interpretation of data; Robert W Morton, Writing - review and editing, Analysis and interpretation of data; Guillaume Paré, Conceptualization, Data curation, Supervision, Funding acquisition, Methodology, Project administration, Writing - review and editing

### Author ORCIDs

Pedrum Mohammadi-Shemirani https://orcid.org/0000-0001-6740-7858
Robert W Morton http://orcid.org/0000-0003-3099-4167
Guillaume Paré https://orcid.org/0000-0002-6795-4760

## Ethics

Human subjects: UK Biobank received ethical approval from the North West Multi-Centre Research Ethics Committee (REC reference: 11/NW/0382). This research was conducted using the UK Biobank under Application Number 15255.

## Decision letter and Author response

Decision letter https://doi.org/10.7554/eLife.58914.sa1
Author response https://doi.org/10.7554/eLife.58914.sa2

## Additional files

### Supplementary files

• Supplementary file 1. Supplementary Tables. Table 1. Characteristics at recruitment for study population of males from UK Biobank cohort study Table 2. Independent genetic variants associated with calculated free testosterone (CFT) at genome-wide significance (p<5×10-8) and not associated with sex hormone-binding globulin in males Table 3. Results of Mendelian randomization analysis using Egger regression for 22 a priori outcomes relevant to testosterone treatment Table 4. Results of Mendelian randomization analysis using MR-RAPS for effect of CFT on 22 a priori outcomes relevant to testosterone treatment Table 5. Results of Mendelian randomization analysis using MR-PRESSO for effect of CFT on 22 a priori outcomes relevant to testosterone treatment Table 6. Associations of genetically-predicted CFT for 439 health outcomes across the human phenome excluding individuals on antihypertensive medication Table 7. Associations of genetically-predicted CFT for 439 health outcomes across the human phenome excluding individuals on cholesterol-lowering medication Table 8. Independent genetic variants associated with total testosterone at genome-wide significance (p<5×10-8) and not associated with sex hormone-binding globulin in 175,421 males from UK Biobank Table 9. All Mendelian randomization analyses of total testosterone on 22 a priori outcomes Table 10. Associations of genetically-predicted total testosterone for 439 health outcomes across the human phenome. Table 11. Definitions for 22 health outcomes with suspected relevance with testosterone treatment Table 12. Definitions for 439 phenome-wide health outcomes *Figure 1*. Screenshot of options shown to male UK Biobank participants for selection of hair/baldness pattern.

• Transparent reporting form

## Data availability

Individual-level data cannot be provided, but it is available to all researchers by application to the UK Biobank. Summary-level GWAS data will be returned to the UK Biobank Access Team for use by other researchers. All MR results and genome-wide significant SNPs have been provided in Supplementary Tables 4 to 12 in Supplementary file 1.

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
