## [Decision Letter]

**Acceptance summary:**

We appreciate the key role of Mendelian randomization analyses in assessing the wide spectrum of testosterone related outcomes in men that you explore in this paper. The approach well complements rigorous randomized trials and provides important information that such large data sets can inform on. Given the increasing use of testosterone in older men in many countries, these data are important in both confirming and extending what trials have shown and in highlighting the lack of effects of lifelong testosterone levels on cardiovascular, cognitive and metabolic outcomes.

**Decision letter after peer review:**

Thank you for submitting your article "Effects of lifelong testosterone exposure on health and disease: a Mendelian randomization study" for consideration by *eLife*. Your article has been reviewed by three peer reviewers, and the evaluation has been overseen by a Reviewing Editor and Eduardo Franco as the Senior Editor.

The reviewers have discussed the reviews with one another and the Reviewing Editor has drafted this decision to help you prepare a revised submission.

This paper provides a large amount of new data on the role of testosterone levels in metabolic outcomes. It provides support for insights gained from trials as well as new insights in especially controversial areas.

Please direct your revisions to the matters discussed by the reviewers as below and consider the additional analyses that are requested as the editors believe addressing these issues will strengthen your paper.

Reviewer #1:

The long-term benefits and risks of testosterone are incompletely understood. Recent randomized trials, especially the TTrials, have taught us a great deal about the efficacy of testosterone replacement therapy, the long-term risks of MACE, prostate cancer, and diabetes remain unclear. Although randomized controlled trials remain the gold standard, MR studies, such as these, can provide useful complementary data.

In this manuscript, the authors performed Mendelian randomization analyses to infer phenome-wide effects of free testosterone on large number of outcomes in male participants of the UK Biobank study. The analyses yielded several very interesting findings, including the positive association of genetically-determined free testosterone levels with increased bone mineral density, and decreased body fat; decreased HDL, and increased risks of prostate cancer, androgenic alopecia, hypertension; and some other phenotypes. The analyses included individual-level data from a large sample from the UK Biobank. Some of the outcomes that were analyzed are especially important: diabetes, prostate cancer, venous thrombosis, myocardial infarction, and stroke because testosterone's effects on these outcomes have remain unclear from the RCTs published so far. The authors performed several sensitivity analyses to ensure that the effects were not driven by any single variant. Additional, Egger regression and MR-PRESSO were performed to detect and correct for potential pleiotropy. For prostate cancer, a two-sample MR analysis was performed using data from the PRACTICAL Consortium and the UK Biobank.

A small number of MR studies have previously investigated the effects of total testosterone on lipids, bone mineral density, and CVD risk; many of them have included relatively smaller samples. This study represents a comprehensive effort to estimate the relation of genetic loci associated with calculated free testosterone levels with a large number of outcomes.

Another recently published MR study by Ruth et al. in Nature Medicine (cited in this paper) reported that genetically determined testosterone levels were associated with sexually dimorphic effects on diabetes risk (Ruth et al., 2020). Although some of the conditions and phenotypes analyzed by Ruth et al. are similar to those reported in this manuscript, the current manuscript includes additional analyses that were not evaluated by Ruth et al. Furthermore, some of the findings differ from those reported by Ruth et al. Therefore, the manuscript includes some important novel information beyond that which was reported by Ruth et al.

A very large body of data are presented, representing a huge amount of work. Thus, the information presented in this manuscript represents an important addition to the extant literature on this topic.

Specific comments:

1) The ascertainment of outcomes and diagnoses using electronic medical records has some inherent problems. These problems are greater for ascertaining some outcomes such as dementia, Alzheimer's dementia, depression, BPH, etc because of the lack of rigorously defined pre-specified diagnostic criteria in clinical practice, lack of prospective adjudication, and nonuniform application of diagnostic codes by clinicians in practice. Many types of lower urinary tract symptoms get coded as BPH. These limitations should be acknowledged in the Discussion.

2) A substantial fraction of adult men (1.5 to 1.7% in the US) are being treated with testosterone. Were testosterone-treated men excluded from the analyses? This is important because testosterone treatment. could confound the analyses. Also, men with genetically-determined low testosterone level are at increased risk of getting treated with testosterone. Therefore, it would be important to know what fraction of people in the UK Biobank data were treated with testosterone and whether they were excluded from the analyses.

3) Total testosterone levels were measured using a platform-based immunoassay that are well known to lack accuracy, especially in the low range. More importantly, free testosterone levels were calculated using an equation that is based on a linear model of testosterone binding to SHBG, which has been shown to be erroneous. The measurement problems are perhaps reflected in the fact that the average calculated free testosterone level in the sample (0.21 nmol/L) is substantially lower than the mean free testosterone levels determined by equilibrium dialysis, the reference method. The reviewer recognizes that the authors had no choice but to use the data that were available in the UK Biobank. But acknowledging this limitation in the Discussion would be important.

4) Also, the basic characteristics (LLOQ, precision and accuracy, specificity, analytical range) of the assays should be provided.

5) How much of the variation in calculated free T levels was explained by the genetic loci that were associated with CFT levels? The Manhattan plot in Figure 1—figure supplement 3 shows the distribution of p-values from genome-wide association study of calculated free testosterone after exclusion of SHBG-associated variants based on chromosomal location. This figure contains really important data. Although the GWAS of total and free T levels have been published, it would be very useful to include the information on these loci and whether any new loci were discovered.

6) Figure 1—figure supplement 1. The units for SHBG are in log units which would be difficult for the readers to comprehend; changing the units to nmol/L would make it easier to get a sense of the distribution of values.

7) The authors found significant associations with some really clinically important outcomes, such as prostate cancer, prostate cancer, androgenic alopecia, and hypertension. Some discussion of the effect size and meaningfulness of the observed effect would be valuable in putting these observations in clinical context.

8) Some of the findings of the analyses, especially on diabetes and prostate cancer risk, differ from those reported by Ruth et al. The authors should comment on why the findings differ in the two sets of analyses that used the same body of UK Biobank data.

9) It is stated that the methods for outcome ascertainment are included in a table in a supplementary file. Criteria for some of the outcomes are provided (e.g., alopecia, depression); I may have missed it, but I did not find the criteria for outcome ascertainment that were used in the definition of a number of other outcomes (e.g., dementia, diabetes, BPH, prostate cancer, etc.).

Reviewer #2:

This is a very interesting and valuable study using a mendelian randomization approach to infer (with the appropriate caveats) causal effects of genetically determined serum testosterone on a variety of phenotypes considered to be androgen sensitive. Strengths include the large cohort (albeit limited to white UK men), and the careful analyses conducted. Some outcomes are expected, others perhaps less so, and may represent chance finding.

Comments to the authors:

1) Given that one of the view aspects that all testosterone guidelines agree on is that total testosterone is the principal measurement to confirm a clinical diagnosis of androgen deficiency, it would be interesting to present results according to total testosterone-or at least defend decision to not do so; while the “free hormone hypothesis” is supported by some studies, not all experts agree on this, as the evidence is not definitive.

2) It is not clear whether testosterone (and SHGB) were measured only once, and if so whether they were drawn in the morning in the fasted state. This is important given the diurnal variability of testosterone measurements, effects of food intake and day to day variability. Moreover, immunoassay for testosterone can be imprecise, especially at the lower range. All these factors may have limited the precision of the GWAS.

3) Interestingly, the average CFT was 0.21 nmol/L in the population (Results first paragraph) is, in the context of sexual symptoms, below the cutoff for diagnosing "Late onset hypogonadism" DOI: 10.1056/NEJMoa0911101. It is not clear whether serum testosterone was measured across the population or only in men in whom it was clinically indicated; either way the low average is surprising and requires further explanation.

4) Discussion paragraph one: "the predicted 1.5-fold increase...observed after initiation of testosterone supplementations", please clarify where these data are from.

5) Discussion paragraph five: the dichotomy between “lifestyle” and “clinical” perspective is a little forced-please rephrase. The clinical approach to testosterone treatment involves weighing benefits (e.g. body composition that may be metabolically favourable or on BMD that may (or may not) reduce fracture risk) against risks. As a matter of course while testosterone replacement in men with organic hypogonadism is undisputed, the role of testosterone treatment for symptomatic men with age-related decline in testosterone remains uncertain. I suggest to avoid the term “supplementation” as it infers correcting a clear hormone deficiency state and instead use the more neutral term “treatment” which acknowledges the possibility that treatment may be pharmacological instead of replacement.

6) Abstract: "MR suggests lifestyle benefits" this is not clear please rephrase.

Reviewer #3:

This manuscript applies Mendelian randomization to investigate potential causal effects of testosterone on biomarkers and health outcomes. Although the statistical methods used by the authors are generally appropriate, there is still some room for improvement (see the comments below). I hope the authors can address them in a revision.

1) I think the Mendelian randomization results will become a lot stronger if the authors can compare the estimated effects on the 22 a priori outcomes with the existing results from RCT (for example, using a scatterplot of MR effects versus RCT effects, with standard error bars in both directions). This will not only reveal whether there is any systematic bias of the MR design/method for testosterone but also how much the "lifelong" effect estimated by MR is larger than short term effect estimated by RCT.

2) Given all the methodological developments for MR, I am surprised to see that the authors chose to report the results of inverse variance weighting (IVW) estimator instead of the other more robust methods. IVW is only valid in the ideal theoretical setting, which is rarely the case for empirical applications. For example, in Figure 3—figure supplement 3 it is very clear that there are a few negative outliers and the IVW slope (or MR-Egger slope) seems to underestimate the positive effect suggested by the majority of the SNPs. This issue can be addressed by MR-PRESSO, but an even better alternative is MR-RAPS that handles outliers, overdispersion, and the many weak instrument asymptotic variance. Related software resources and discussion can be found in the links below:

https://github.com/qingyuanzhao/mr.raps

https://doi.org/10.1093/ije/dyz142

3) A statistical issue unaddressed by the authors is the winner's curse in selecting the genetic instruments. This happens if the same GWAS is used to both select instruments and make statistical inference. In general, the winner's curse biases the point estimator towards 0 in two-sample MR, but that bias can be more complicated when compounded with other issues like outliers. The winner's curse can be eliminated by using a three-sample MR design, in which a separate dataset is used to select instruments; see the paper in the second link above. If this is not possible, the best alternative I know is to use a very strict significance threshold for instrument selection (which the authors have already done) and acknowledge the potential bias from winner's curse in the discussion.

[Editors' note: further revisions were suggested prior to acceptance, as described below.]

Thank you for resubmitting your work entitled "Effects of lifelong testosterone exposure on health and disease using Mendelian randomization" for further consideration by *eLife*. Your revised article has been evaluated by the editors after consultation with the original reviewers.

The manuscript has been improved but there are two remaining issues that need to be addressed before full acceptance is made, as outlined below:

One of the reviewers, who is enthusiastic about this paper, strongly requests that you make two additional analyses and/or explanations. That reviewer, an expert in the testosterone field, states:

1) One issue that I am still puzzled about is the difference in the findings from the data reported by Ruth et al. with respect to the association between genetically determined free testosterone and diabetes risk. The authors list "fewer cases of T2D leading to inadequate statistical power or other differences in the populations used for T2D analysis." If the same database of UK Biobank was used in both the analyses, why would there be a difference in the number of T2D cases or in the study population?

2) I also continue to be concerned about the calculated free T concentrations that are substantially lower than those described previously in community-dwelling men. I recognize that these are the numbers that the UK Biobank has provided, but it would be worth re-checking the calculations to make sure there is no inadvertent systematic error in computation.

Comment: The authors have been very diligent in providing a very large body of data in the supplementary tables and figures. I do not recommend inclusion of any additional data beyond what is included in the current manuscript.

---

## [Author Response]

Reviewer #1:[…]Specific comments1) The ascertainment of outcomes and diagnoses using electronic medical records has some inherent problems. These problems are greater for ascertaining some outcomes such as dementia, Alzheimer's dementia, depression, BPH, etc because of the lack of rigorously defined pre-specified diagnostic criteria in clinical practice, lack of prospective adjudication, and nonuniform application of diagnostic codes by clinicians in practice. Many types of lower urinary tract symptoms get coded as BPH. These limitations should be acknowledged in the Discussion.

We agree with the reviewer’s comments regarding the limitations of using data from electronic medical records. Unfortunately, these issues are unavoidable with the data at our disposal, and as such, we have explicitly highlighted the potential issues in the Discussion, “Relatedly, an inherent limitation for outcomes ascertained using linked electronic medical records is a lack of adjudication and consistent application of codes in clinical practice.”.

2) A substantial fraction of adult men (1.5 to 1.7% in the US) are being treated with testosterone. Were testosterone-treated men excluded from the analyses? This is important because testosterone treatment. could confound the analyses. Also, men with genetically-determined low testosterone level are at increased risk of getting treated with testosterone. Therefore, it would be important to know what fraction of people in the UK Biobank data were treated with testosterone and whether they were excluded from the analyses.

This is an important consideration for a cohort study such as the UK Biobank. For our analyses, we did exclude participants that self-reported taking androgen medication based on field ID 20003. This exclusion was previously described for the genome-wide association study, but we have now clarified the exclusion applied to the entire study by moving the sentence earlier in the revised Materials and methods. The fraction of participants excluded may have been less than 1.5-1.7% due to differences in rates of testosterone prescription between the United Kingdom and United States of America (Handelsman DJ, 2013).

3) Total testosterone levels were measured using a platform-based immunoassay that are well known to lack accuracy, especially in the low range. More importantly, free testosterone levels were calculated using an equation that is based on a linear model of testosterone binding to SHBG, which has been shown to be erroneous. The measurement problems are perhaps reflected in the fact that the average calculated free testosterone level in the sample (0.21 nmol/L) is substantially lower than the mean free testosterone levels determined by equilibrium dialysis, the reference method. The reviewer recognizes that the authors had no choice but to use the data that were available in the UK Biobank. But acknowledging this limitation in the Discussion would be important.

The reviewer brings up an important point regarding the accuracy of immunoassays, and its limitations relative to the gold-standard of equilibrium dialysis. Since this is inherent to the UK Biobank, we have elaborated in the Discussion, “In the UK Biobank, calculated free testosterone levels were below the reference ranges for young healthy individuals, which may be attributable to the older age of the cohort and inherent inaccuracy of immunoassays at lower levels of total testosterone.”.

4) Also, the basic characteristics (LLOQ, precision and accuracy, specificity, analytical range) of the assays should be provided.

In line with the previous comment, we have included the analytical characteristics of the assay as provided by the UK Biobank in the Materials and methods, “Analytical range for the immunoassays of total testosterone and SHBG were 0.35 to 55.52 and 0.33 to (226-242) nmol/L, respectively. For total testosterone, within-laboratory CV for high, medium, and low concentration quality control samples were 4.15, 3.66, and 8.34%. For SHBG, within-laboratory CV for high, medium, and low concentration quality control samples were 5.22, 5.25, and 5.67%.”.

5) How much of the variation in calculated free T levels was explained by the genetic loci that were associated with CFT levels? The Manhattan plot in Figure 3—figure supplement 3 shows the distribution of p-values from genome-wide association study of calculated free testosterone after exclusion of SHBG-associated variants based on chromosomal location. This figure contains really important data. Although the GWAS of total and free T levels have been published, it would be very useful to include the information on these loci and whether any new loci were discovered.

As per the reviewer’s suggestions, we have clarified the amount of variation explained by the loci associated with CFT at genome-wide significant loci, “Overall chip-based heritability of CFT was estimated at 15 % (95%CI = 14 to 16), while these 93 independent genetic variants associated with CFT explained 3.7% of the total variance of CFT levels in males from the UK Biobank.” . Moreover, the nearest gene(s) to independent GWS loci pictured in Figure 1—figure supplement 3 are annotated in Supplementary file 1—table 2 alongside additional details regarding each genetic variant.

6) Figure 1—figure supplement 1. The units for SHBG are in log units which would be difficult for the readers to comprehend; changing the units to nmol/L would make it easier to get a sense of the distribution of values.

Although the log-transformed values reflect the units and distribution used in our subsequent analyses, we understand that units in nmol/L are more interpretable for readers. As a result, we have added Figure 1—figure supplement 3A to reflect raw values as nmol/L while Figure 1—figure supplement 3B reflects log-transformed values.

7) The authors found significant associations with some really clinically important outcomes, such as prostate cancer, prostate cancer, androgenic alopecia, and hypertension. Some discussion of the effect size and meaningfulness of the observed effect would be valuable in putting these observations in clinical context.

We feel this is an important comment as it allowed us to reflect on better communicating our findings. For all outcomes, we have represented the effect sizes in terms of 0.1 nmol/L increase of free testosterone, which reflects approximate changes observed during initiation of testosterone supplementation. We have emphasized this decision in the Discussion, “All effects are reported in terms of 0.1 nmol/L of CFT to approximate expected effect sizes after initiation of testosterone treatment.” and Figure 3 legend.

8) Some of the findings of the analyses, especially on diabetes and prostate cancer risk, differ from those reported by Ruth et al. The authors should comment on why the findings differ in the two sets of analyses that used the same body of UK Biobank data.

We briefly touched on the differences between our analyses in the Introduction. Ruth et al. similarly found a risk-conferring effect of testosterone on prostate cancer (OR = 1.23 per 1 SD bioavailable testosterone; 95% CI = 1.13 to 1.33). However, we have revised the Discussion to further elaborate on methodological differences that might explain the divergent findings regarding diabetes, “The apparent difference between protective effects of testosterone observed in a previous MR analysis of testosterone and lack of protective effect in our study might be a result of less stringent control for pleiotropic effects of SHBG in the previous study. Given studies have identified associations between SHBG and risk of T2D independent of testosterone and a direct role of SHBG in mediating signaling on target cells, insufficient control for SHBG may lead to residual pleiotropic effects (Vikan et al., 2010) (Lakshman, Bhasin and Araujo, 2010) (Rosner et al., 2010). Other reasons may include genetic variants explaining less variation in testosterone levels in our study, fewer cases of T2D leading to inadequate statistical power to detect weaker effects in our study, or other differences in the populations used for T2D analysis.”.

9) It is stated that the methods for outcome ascertainment are included in a table in a supplementary file. Criteria for some of the outcomes are provided (e.g., alopecia, depression); I may have missed it, but I did not find the criteria for outcome ascertainment that were used in the definition of a number of other outcomes (e.g., dementia, diabetes, BPH, prostate cancer, etc.)

We apologize for any confusion. The definitions for 22 *a priori* outcomes were listed in Supplementary file 1—table 11. These included the field IDs and ICD-10 codes, if applicable, for dementia (row 7), BPH (row 16), prostate cancer (row 22), and type 2 diabetes (row 25). The definitions for 439 phenome-wide outcomes were listed in Supplementary file 1—table 12. We further explain this distinction, “Detailed descriptions and selection criteria are available for all a priori outcomes in Supplementary file 1—table 11, and phenome-wide outcomes and biomarkers in Supplementary file 1—table 12.”, and now provide more details on the definition of outcomes in the revised Materials and methods.

Reviewer #2:[…]Comments to the authors:1) Given that one of the view aspects that all testosterone guidelines agree on is that total testosterone is the principal measurement to confirm a clinical diagnosis of androgen deficiency, it would be interesting to present results according to total testosterone-or at least defend decision to not do so; while the “free hormone hypothesis” is supported by some studies, not all experts agree on this, as the evidence is not definitive.

Although we agree the “free hormone hypothesis” is not definitive, we felt this dataset presented a unique opportunity to explore the effects of free testosterone specifically. Importantly, we repeated our analyses using genetically-predicted total testosterone (Supplementary file 1—tables 9 and 10) and found largely consistent effects with the significant results using genetically-predicted free testosterone. We have added a comment to this effect in the Discussion, “Furthermore, although the free hormone hypothesis is still debated by experts, we found largely consistent effects on outcomes using genetically-predicted free testosterone and total testosterone. […] Indeed, one of the pleiotropic outliers identified by MR-PRESSO was rs9986829, which is located near *DGKB* – a gene associated with glucose homeostasis and type 2 diabetes in multiple cohorts (10.1371/journal.pone.0015542) (10.1038/ng.520).

2) It is not clear whether testosterone (and SHGB) were measured only once, and if so whether they were drawn in the morning in the fasted state. This is important given the diurnal variability of testosterone measurements, effects of food intake and day to day variability. Moreover, immunoassay for testosterone can be imprecise, especially at the lower range. All these factors may have limited the precision of the GWAS.

We thank the reviewer for bringing this to our attention. We have clarified that testosterone was measured only once in the Materials and methods and acknowledge the limitations associated with this source of variability in the Discussion, “Additional sources of variability introduced into the total testosterone measurements include differences in fasting times, diets, and time of day at which blood was drawn from participants. Nevertheless, genetic variants associated with testosterone consistently replicated known effects of testosterone on established outcomes, such as body fat, body fat-free mass, and haematocrit.”. Likewise, we have acknowledged the inherent limitations of the immunoassay in the revised Discussion.

3) Interestingly, the average CFT was 0.21 nmol/L in the population (Results first paragraph) is, in the context of sexual symptoms, below the cutoff for diagnosing "Late onset hypogonadism" DOI: 10.1056/NEJMoa0911101. It is not clear whether serum testosterone was measured across the population or only in men in whom it was clinically indicated; either way the low average is surprising and requires further explanation.

Testosterone was measured across the population, but the older average age (57 years) in the UK Biobank may explain the lower calculated free testosterone levels relative to reference ranges from healthy adult populations. Indeed, mean total testosterone levels were similarly low relative to reference ranges, and previous studies utilizing the UK Biobank have reported comparable levels for both free and total testosterone (Peila, Arthur and Rohan, 2020) (Petermann-Rocha et al., 2020). This is commented on in the Discussion, “In the UK Biobank, calculated free testosterone levels were below the reference ranges for young healthy individuals, which may be attributable to the older age of the cohort and inherent inaccuracy of immunoassays at lower levels of total testosterone. Total testosterone levels are similarly low relative to reference ranges and comparable to previous studies in the UK Biobank.”.

4) Discussion paragraph one: "the predicted 1.5-fold increase...observed after initiation of testosterone supplementations", please clarify where these data are from.5) Discussion paragraph five: the dichotomy between “lifestyle” and “clinical” perspective is a little forced-please rephrase. The clinical approach to testosterone treatment involves weighing benefits (e.g. body composition that may be metabolically favourable or on BMD that may (or may not) reduce fracture risk) against risks. As a matter of course while testosterone replacement in men with organic hypogonadism is undisputed, the role of testosterone treatment for symptomatic men with age-related decline in testosterone remains uncertain. I suggest to avoid the term “supplementation” as it infers correcting a clear hormone deficiency state and instead use the more neutral term “treatment” which acknowledges the possibility that treatment may be pharmacological instead of replacement.

The intent behind the “lifestyle” and “clinical” distinction was to classify and improve interpretability of our findings, but we understand the reviewer’s concerns. Consequently, we replaced all instances of “testosterone supplementation” with “testosterone treatment” and reworded our statement to include all benefits and adverse effects as a whole, “Beneficial effects on body composition, sexual function, hematocrit, and BMD should be weighed against detrimental effects on androgenic alopecia, prostate cancer, hypertension and spinal stenosis, and no detectable beneficial effects on other major clinical endpoints.”.

6) Abstract: "MR suggests lifestyle benefits" this is not clear please rephrase.

In the same manner as the previous comment, we have removed any reference to “lifestyle” from the manuscript.

Reviewer #3:[…]1) I think the Mendelian randomization results will become a lot stronger if the authors can compare the estimated effects on the 22 a priori outcomes with the existing results from RCT (for example, using a scatterplot of MR effects versus RCT effects, with standard error bars in both directions). This will not only reveal whether there is any systematic bias of the MR design/method for testosterone but also how much the "lifelong" effect estimated by MR is larger than short term effect estimated by RCT.

We thank the reviewer for this insight. We weren’t able to find RCTs of testosterone therapy for all significant outcomes in the MR analyses, but we have included a plot comparing RCT versus MR effect estimates for available outcomes in Figure 3. Data sources are referenced in the Materials and methods, “Finally, for significant outcomes, we compared estimated effect sizes from this MR study with reported effect sizes from random controlled trials of testosterone therapy, where possible…”, and we have expanded on results in the Discussion, “Most of the estimates from MR analyses were comparable with effect sizes from RCTs (Figure 3). There was only significant heterogeneity between effects on BMD for MR and RCT, but it is difficult to make direct comparisons due to variable change in testosterone levels after administration of testosterone in each RCT, different methods and anatomical sites of BMD estimation, and differences between short-term effects in RCTs relative to lifelong effects in MR.”.

2) Given all the methodological developments for MR, I am surprised to see that the authors chose to report the results of inverse variance weighting (IVW) estimator instead of the other more robust methods. IVW is only valid in the ideal theoretical setting, which is rarely the case for empirical applications. For example, in Figure 3—figure supplement 3 it is very clear that there are a few negative outliers and the IVW slope (or MR-Egger slope) seems to underestimate the positive effect suggested by the majority of the SNPs. This issue can be addressed by MR-PRESSO, but an even better alternative is MR-RAPS that handles outliers, overdispersion, and the many weak instrument asymptotic variance. Related software resources and discussion can be found in the links below:https://github.com/qingyuanzhao/mr.rapshttps://doi.org/10.1093/ije/dyz142

We thank the reviewer for bringing this to our attention. As a sensitivity analysis, MR-RAPS has been conducted for all 22 *a priori* outcomes as described in the revised Materials and methods “As a sensitivity analysis robust to idiosyncratic pleiotropy and weak instrument bias, MR-RAPS (Robust Adjusted Profile Score) was conducted using overdispersion and Tukey’s loss function.”, Results, “Results using MR-RAPS were consistent with IVW regression method for all significant outcomes.”, and Supplementary file 1—table 4 and table 10.

3) A statistical issue unaddressed by the authors is the winner's curse in selecting the genetic instruments. This happens if the same GWAS is used to both select instruments and make statistical inference. In general, the winner's curse biases the point estimator towards 0 in two-sample MR, but that bias can be more complicated when compounded with other issues like outliers. The winner's curse can be eliminated by using a three-sample MR design, in which a separate dataset is used to select instruments; see the paper in the second link above. If this is not possible, the best alternative I know is to use a very strict significance threshold for instrument selection (which the authors have already done) and acknowledge the potential bias from winner's curse in the discussion.

We agree with the reviewer’s remarks regarding bias due to the winner’s curse phenomenon. Due to limited data availability, employing a three-sample MR design would be challenging as we don’t have access to a sufficiently well-powered third study to select instruments. As a result, we have acknowledged this source of bias in the Discussion, “Although a stringent p-value threshold was selected for genetic variants, the winner’s curse phenomenon may still bias effect sizes due to the same sample being used to select genetic variants and estimate effect sizes on testosterone.”.

[Editors' note: further revisions were suggested prior to acceptance, as described below.]

One of the reviewers, who is enthusiastic about this paper, strongly requests that you make two additional analyses and/or explanations. That reviewer, an expert in the testosterone field, states:1) One issue that I am still puzzled about is the difference in the findings from the data reported by Ruth et al. with respect to the association between genetically determined free testosterone and diabetes risk. The authors list "fewer cases of T2D leading to inadequate statistical power or other differences in the populations used for T2D analysis." If the same database of UK Biobank was used in both the analyses, why would there be a difference in the number of T2D cases or in the study population?

We understand the confusion and hope to clarify. Although both studies used the UK Biobank as the source of testosterone data, there were differences in the source of genetic estimates for outcomes, such as type 2 diabetes. Our study employed a one-sample Mendelian randomization design deriving estimates from the UK Biobank, whereas *Ruth et al.* employed a two-sample Mendelian randomization design deriving estimates from male-specific GWAS summary statistics from the DIAGRAM consortium. The DIAGRAM consortium contains 34,990 cases of type 2 diabetes, whereas UK Biobank contains 11,079 cases. However, sex-specific results of the DIAGRAM consortium are unpublished and unavailable to the public at this time. Despite constraints on data availability, we attempted to approximate analyses performed by *Ruth et al.* in the context of our study design by combining SNPs associated with bioavailable testosterone from their study with the same T2D cases from UK Biobank used in our analysis. Results are consistent with our study showing no effect on type 2 diabetes across IVW (OR=0.97 (95%CI=0.86 to 1.08); p=0.55), MR-PRESSO (OR=1.02 (95%CI=0.93 to 1.12); p=0.65), and MR-RAPS (OR=0.99 (95%CI=0.89 to 1.10); p=0.88). Therefore, in addition to the other noted differences, such as more robust control for pleiotropic effects of SHBG in our study, this could explain the difference in the association with type 2 diabetes.

We’ve amended the manuscript to better clarify this distinction in the Discussion, “The apparent difference between protective effects of testosterone observed in a previous MR analysis of testosterone and lack of protective effect in our study might be a result of less stringent control for pleiotropic effects of SHBG in the previous study. […] Other reasons may include genetic variants explaining less variation in testosterone levels in our study, fewer cases of T2D leading to inadequate statistical power to detect weaker effects in our study, or other differences between the populations of the UK Biobank in our study and DIAGRAM consortium used by Ruth et al.”.

2) I also continue to be concerned about the calculated free T concentrations that are substantially lower than those described previously in community-dwelling men. I recognize that these are the numbers that the UK Biobank has provided, but it would be worth re-checking the calculations to make sure there is no inadvertent systematic error in computation.

We appreciate the reviewer’s insight into this abnormality. We share the curiosity regarding the apparently lower levels of testosterone in the UK Biobank population, so we rechecked our calculations against the Vermeulen equation and found no error. As we comment on in the Discussion, the low levels may be attributable to the older average age in the UK Biobank and/or use of immunoassays in the measurement of sex hormones. Indeed, average total testosterone is similarly low (11.9 nmol/L or 343 ng/dL) in the UK Biobank relative to reference ranges (Bhasin et al., 2011) (Travison et al., 2017). Most importantly, the free testosterone levels calculated by our group are unlikely to represent a computational error as they are comparable to levels reported by other independent groups using UK Biobank data in both peer-reviewed literature (Peila, Arthur and Rohan, 2020) (Yeap et al., 2020) and preprint (Watts et al., 2020) (Fan et al., 2020).